# The Application of a Self-Organizing Model for the Estimation of Crop Water Stress Index (CWSI) in Soybean with Different Watering Levels

Angela Anda *, Brigitta Simon-Gáspár and Gábor Soós

Georgikon Campus, Hungarian University of Agriculture and Life Sciences, P.O. Box 71,
H-8361 Keszthely, Hungary; Simon.Gaspar.Brigitta@uni-mate.hu (B.S.-G.); Soos.Gabor@uni-mate.hu (G.S.)
* Correspondence: Anda.Angela@uni-mate.hu; Tel.: +36-83-545-149

**Abstract:** A field experiment was conducted with soybean to observe evapotranspiration (ET) and crop water stress index (CWSI) with three watering levels at Keszthely, Hungary, during the growing seasons 2017–2020. The three different watering levels were rainfed, unlimited, and water stress in flowering. Traditional and converted evapotranspirometers documented water stress levels in two soybean varieties (Sinara, Sigalia), with differing water demands. ET totals with no significant differences between varieties varied from 291.9 to 694.9 mm in dry, and from 205.5 to 615.6 mm in wet seasons. Theoretical CWSI, $CWSI_t$ was computed using the method of Jackson. One of the seasons, the wet 2020 had to be excluded from the $CWSI_t$ analysis because of uncertain canopy temperature, $T_c$ data. Seasonal mean $CWSI_t$ and $T_c$ were inversely related to water use efficiency. An unsupervised Kohonen self-organizing map (K-SOM) was developed to predict the CWSI, $CWSI_p$ based on easily accessible meteorological variables and $T_c$. In the prediction, the $CWSI_p$ of three watering levels and two varieties covered a wide range of index values. The results suggest that $CWSI_p$ modelling with the minimum amount of input data provided opportunity for reliable $CWSI_p$ predictions in every water treatment ($R^2 = 0.935$–$0.953$; RMSE = 0.033–0.068 mm, MAE = 0.026–0.158, NSE = 0.336–0.901, SI = 0.095–0.182) that could be useful in water stress management of soybean. However, highly variable weather conditions in the mild continental climate of Hungary might limit the potential of CWSI application. The results in the study suggest that a less than 450 mm seasonal precipitation caused yield reduction. Therefore, a 100–160 mm additional water use could be recommended during the dry growing seasons of the country. The 150 year-long local meteorological data indicated that 6 growing seasons out of 10 are short of precipitation in rainfed soybean.

**Keywords:** CWSI; evapotranspiration; K-SOM analysis; WUE; soybean; water stress

## 1. Introduction

Water stress seems likely to occur more often in the near future, leading to a real challenge for global food supply [1]. Water stress in deficit irrigation treatments resulted in lower soybean yield as compared to a fully irrigated one [2]. Reports have shown that during flowering and pod filling stages soybean plants were extremely sensitive to drought [3] causing flower fall, a reduced number of pods and decreased seed size. Water stress imposed at the reproductive stage of soybean increased the biomass partitioning to stems and roots, while partitioning to leaves and seed was drastically reduced on the area of the Mississippi State University [4]. The reduced available water resulted in stomata closure that arrested photosynthesis, adversely influencing different physiological and biochemical processes in soybean [5]. The above authors observed a significant reduction in soybean yield, up to 40%, at half field capacity. As soybean was considered a drought-sensitive crop [6], soybean yield enhancement requires selection of varieties even under variable climate conditions. Selecting new varieties with high grain yield but low water use

may provide a new way to improve soybean yield performance even under water scarce conditions [7].

Among many reports with varying complexity in the literature, including soil water balance, residual energy budget, Bowen ratio modelling, lysimeters and eddy-covariance, to develop accurate ET estimations for irrigation management purposes is strongly advised [8,9].

Enough available soil water intensifies the transpiration of leaves, producing a cooling effect, which reduces the canopy temperature, $T_c$ lower than that of the ambient air one, $T_{ac}$ [10]. When the soil moisture becomes limited, to decrease transpiration water loss, the $T_c$ increases above the $T_{ac}$. Therefore, an index, the canopy and ambient air temperature differential ($T_c$-$T_a$) reflects the self-regulation ability of crops under water stress [11]. Jackson et al. [12] developed the theoretical crop water stress index, $CWSI_t$, as a normalized indicator to account for the varying environmental factors that may affect the ($T_c$-$T_a$); the relationship between water stress and $T_c$. The assumption of Jackson et al. [12] is based on the energy balance of crops. Agam et al. [13] reported the issue in the use of theoretical $CWSI_t$, as it is limited by the necessity of net radiation ($R_n$), aerodynamic resistance ($r_a$) and other model input parameters. In the empirical CWSI development, the sensed $T_c$ is normalized by using two baseline temperatures (upper limit: non-transpiring canopy; lower limit: non-water-stressed baseline) [10]. To quantify the empirical CWSI, the linear relationship for the temperature differential and vapor pressure deficit, VPD was established. It has also been shown that $T_c$ is crop growth stage- and climate zone-dependent, in which the crop is being grown [14]. Maes and Steppe [15] and Jones [16] called attention to the required "stable" weather conditions to measure $T_c$ that is seldom encountered under humid climate conditions. Despite these concerns, the CWSI is widely used for different (biotic and abiotic) stress detections, irrigation timing as a quantitative parameter to detect the influence of drought on crops.

The K-SOM (Kohonen self-organizing maps), based on artificial neural network algorithms, is considered a simple tool for the organization of complex data according to their similarities, providing pattern recognition [17]. Neurons are put in nodes of the lattices that are selectively tuned to input patterns of a competitive learning process [18]. The unsupervised K-SOM do not have specific input or output variables because all variables in the input vector are also found in each unit of the output layer [19]. Thus, K-SOM can extract useful information even from noisy data [20]. Kumar et al. [21] applied K-SOM to approach empirical CWSI in Indian mustard to eliminate the need to obtain the base temperatures required to calculate the empirical CWSI, which could otherwise be complicated.

From the early application of K-SOM using unsupervised algorithms, where data were easily interpreted, and the clustering helped in identifying similarities in the dataset [19], the SOM has also been advanced in prediction purposes even for water resource modelling [21]. In this paper, the areas of applications in the estimation of soybean CWSI are highlighted.

Soybean-water relation in the study may offer information for breeding programs in identifying improved water stress tolerant varieties. Therefore, soybean yield enhancement requires selection of tolerant and compatible cultivars in dry climate and low water supplies. Enhancement of watering, WUE and soybean performance under water stress should be a primary purpose for soybean breeders, in order to improve and stabilize seed yield. This work contributes to mitigating water scarcity by providing new information about irrigation timing for more successful drought management.

The study aims were to:

(i) quantify the soybean ET and WUE with three watering levels (non-limited, water stressed and rainfed)
(ii) analyze two soybean varieties differing in their water demands
(iii) control the applicability of theoretical $CWSI_t$ under highly variable weather conditions of Hungary

(iv)  test the K-SOM model in the $CWSI_p$ prediction by applying easily accessible meteorological and crop variables ($T_a$, $RH_c$ and $T_c$).

Due to difficult access of meteorological elements and parameters used in computing theoretical $CWSI_t$, only the $T_{ac}$, $T_c$ and $RH_c$ were included in the K-SOM $CWSI_p$ estimation. These variables are easily accessible to users at any meteorological station. The major drawback of applying CWSI is the restriction of making the required measurements just around solar noon under clear-sky and calm weather conditions. The $CWSI_p$ visualized and estimated by the K-SOM model seemed to be a rapid and robust solution for developing a water-saving irrigation scheduling method. Although the biological meaning of crops may be certain using K-SOM-projected $CWSI_p$ only. K-SOM-projected $CWSI_p$ may provide an alternative to other CWSI estimations requiring a large amount of input data and computation. According to the recommendation of Kumar et al. [20], the applicability of K-SOM to predict soybean $CWSI_p$ in Hungarian temperate climate was accomplished. This study is a further development in soybean $CWSI_t$ completed with K-SOM-based index estimation and the short-term publications previously reported by the authors [22,23]. This work may provide valuable information for international readers in the context of climate change, such as increasing drought occurrence. These results could help in preparing mitigation of future challenges related to missing PR.

## 2. Materials and Methods

### 2.1. Site Description, Agronomic Procedures, and Meteorological Observations

The four-season (2017–2020) field experiments were conducted on studying soybean (*Glycine max* (L.) Merr.)-water relation, at the Agrometeorological Research Station of Keszthely, ARS (latitude: 46°44′ N, longitude: 17°14′ E, elevation: 124 m above sea level), a part of the Hungarian Meteorological Observational Network, run under the provision of the World Meteorological Organization [24]. The climate of the region is mild continental (Cfb) with warm, dry summers and fairly cold winters according to the Köppen-Geiger classification [25], with seasonal mean Ta of 16.9 °C and seasonal PR sum of 384.3 mm (climate norm between 1971 and 2000). Meteorological data sets were collected from VAISALA automatic climate station of QLC-50 type (Vaisala, Helsinki, Finland) equipped with a CM-3 pyranometer (Kipp & Zonen Corp., Delft, The Netherlands). The sensors, except the anemometer, were placed at a standard height of 2 m above the ground surface in the meteorological garden of the ARS. The height of the anemometer was at 10.5 m. The wind speed was extrapolated to 2 m height in calculation of FAO-56 Penman-Monteith reference evapotranspiration, $ET_0$. Description of the computation can be read in Soós and Anda [26]. In addition to standard meteorological observation, which data were used in local weather and season characterization, air temperature, $T_{ac}$ and $RH_c$ were also measured at about 1.5 m above the canopies by combined temperature-humidity sensors (HP472AC combined probe, HD 2101.2, Delta OHM, Padova, Italy) with a log interval of 6 s, at the same time as $T_c$ readings were taken. Microsoft Office Excel 2010 was used to process the data. These meteorological variables were used in theoretical $CWSI_t$ computation.

The dominant soil type was classified as a clay loam, Haplic cambisol (FAO 2006) with a mean bulk density of 1150 kg m$^{-3}$ in the top 1 m of the profile. Plant available water holding capacity to 1 m soil depth was 273 mm m$^{-1}$. The pots of the evapotranspirometers were also filled with the same soil from the upper layer of an adjacent field. Two indeterminate soybean varieties, Sinara (Sin, water stress tolerant) and Sigalia (Sig, for "normal" water use) were seeded in the end of April (2018) or in May (2017, 2019 and 2020) and, irrespective of season, harvested in the first 10 days of each September. The selection of used varieties was made based on the similarity of yield and crop cycle duration and contrasting water demands. Plant density was 40 plant m$^{-2}$, the inter-row spacing was 0.24 m. The density of harvested population was estimated at about 25,000–30,000 plants ha$^{-1}$. Crop growth and agronomic procedures were controlled following the best management practice for soybean prescribed by local agronomists of the University (Georgikon Campus of Keszthely).

*2.2. ET and WUE*

Daily crop reference evapotranspiration rate, $ET_0$ was computed by FAO-56 Penman-Monteith equation [27]. To get net radiation ($R_n$) short-wave balance was computed by on-site measured global radiation and albedo. The long-wave radiation was also calculated by using the method of Allen et al. [27].

Thornthwaite-Mather type compensation evapotranspirometers provided unlimited water supply, WW for soybean. The metal containers with a volume of 4 $m^3$ (surface area: 4 $m^2$, depth of them: 1 m) were fixed in the soil. To have sufficient fetch in all directions, representative of the watered soybean canopy being studied, the containers were surrounded by daily irrigated soybean (at about 25 m radius). The measured daily ET rates were expressed as a residual member from the water balance equation. The loss in total water was calculated by summing daily evapotranspiration rates. In the water-stressed crops, RO 50% water withdrawal was assured by closing the water supplier tap every second day from R1 (beginning bloom) until R4-R5 (grain-filling stage) [28]. A detailed description of the newly reconstructed evapotranspirometer is in Anda et al. [29]. In the RO, the contribution of PR was excluded by means of mobile rainout shelters (2.5 m long, 4.5 m wide, 2.5 m in height covered with 200 μm thick UV transparent polyethylene film). Three replications provided unlimited and stressed water supplies to each variety. The third water treatment was the rainfed, P. Due to the fixed establishment of the evapotranspirometer, the experimental design was a complete block with three replications. The rainfed field, which was about 0.5 ha in size and lay adjacent to the growing pots of the evapotranspirometers, was divided into two sections with the two soybean varieties sown separately, in an area of 50 m (width) × 60 m (length) for each variety. Detailed information about the agronomic procedures, the experimental layout as well as the evapotranspirometers' conversion can be read in pervious short-term publications of Anda et al. [22,23].

In addition to treatments in the containers of the evapotranspirometer, at maturity, 2 × 2 m subplots were highlighted in the adjacent field to get the seed yield of soybean. Five replicates per variety were included in the analysis. The harvested pods were oven dried at 65 °C for at least 48 h (until constant weight) to obtain pod dry weight. After threshing pods by hand, seed weight was obtained and adjusted to 13% moisture. The process was the same in each evapotranspirometer pot.

To calculate the *WUE* [kg $m^{-3}$] for seed yield (*y*), the ratio of *y* and seasonal *ET* total was applied:

$$WUE \left[ \text{kg m}^{-3} \right] = \frac{y \left[ \text{kg m}^{-2} \right]}{\Sigma ET \, [\text{m}]} \tag{1}$$

*2.3. $CWSI_t$ and $T_c$ Readings*

The $T_c$ readings were sensed with a hand-held infrared thermometer (RANGER II. RTL, RAYTEK, Santa Cruz, CA, USA) which has a 2° field of view (FOW) and detects radiation in the 8–14 μm waveband. The resolution of the thermometer was 0.1 °C. Observations were initiated at flowering (DOY: ~190–195) when the crops reached the 70–80% canopy cover to avoid sensing the temperature of the soil and continued until the crops matured. The thermometer was held about 0.5–1 m above the canopy surface at an oblique angle ($\approx$30–40°) below the horizontal. The thermometer sensor was calibrated by the manufacturer. $T_c$ samples were taken every 2 s, 20–25 readings were averaged for each measurement, and this was repeated 3–5 times at high solar angles (local standard time: 11 a.m.–3.00 p.m.) when the sky was completely cloudless (global radiation between 600–900 W $m^{-2}$) with wind speed less than 2 m $s^{-1}$ above the canopy. According to Pajero and Irmak [30], each observation of $T_c$ was detected from two directions (east and west) and averaged to determine the mean $T_c$ of the actual treatment. The emissivity of the soybean was set at 0.96 [31].

Measured $T_c$ data were the basis in the $CWSI_t$ computation following the method of [12]. $CWSI_t$ equals the ratio of measured evapotranspiration (ET) to reference evapotranspiration ($ET_0$):

$$CWSI_t = 1 - \frac{ET}{ET_0} = \frac{\gamma(1 + r_c/r_a) - \gamma}{\Delta + \gamma(1 + r_c/r_a)} \quad (2)$$

where $r_c$ and $r_a$ are canopy and aerodynamic resistances (s m$^{-1}$), $\gamma$ is a psychrometric constant [hPa K$^{-1}$], and $\Delta$ is the slope of saturated vapor pressure-temperature relation [hPa K$^{-1}$].

To calculate the resistance ratio $r_c/r_a$ the following equation was applied:

$$\frac{r_c}{r_a} = \frac{\gamma r_a R_n / (\rho c_p) - (T_c - T_{ac})(\Delta + \gamma) - (e_s(T_c) - e)}{\gamma[(T_c - T_{ac}) - r_a R_n / (\rho c_p)]} \quad (3)$$

where $e_s(T_c) - e$ is the difference in saturation and actual vapor concentrations of air (hPa), $c_p$ is heat capacity of air in constant pressure (J kg$^{-1}$ K$^{-1}$), $p$ is air density (kg m$^{-3}$), and $R_n$ is net radiation (W m$^{-2}$). The ra was computed by the method of Thom and Oliver [32].

### 2.4. Kohonen Self-Organizing Maps (K-SOM)

The self-organizing map (SOM) is a type of neural network used for the visualization and interpretation of large high-dimensional data sets [33]. The essence of SOM's visualization technique is to reduce dimensions of data and projects into a two-dimensional map. SOM typically has two aspects: input data, and output data sets or map usually laid out as a two-dimensional hexagonal lattice [34].

A SOM algorithm is usually implemented in three steps:

—  selections of a specified number of neurons and random initialization of the weights of the components for each neuron.
—  performing iterative training where the nodes are adjusted in response to a set of training vectors, so that the nodes approximately minimize an integrated distance criterion.
—  map visualization where each node's reference vector is projected in some fashion to a lower dimensional space and plotted as a map [35,36].

During training phase, input data is presented to the algorithm and a SOM map is created. Units close to each other on the map have similar input patterns.

After normalization, the training phase generated and updated the weight vectors of the map. Model validation was important to determine the generalizability of K-SOM. The BMU for each input vector was then determined during validation to predict missing CWSI values (Figure 1). After the CWSI values were extracted from the BMU's, they were compared with their computed values to evaluate the performance of the K-SOM predictions.

### 2.5. Statistics

The effect of the treatment, variety and season on the ET rate, and the WUE was analyzed with three-way ANOVA. The starting model was the full model with all main effects and all two- and three-way interactions. The pairwise comparisons of the eight water-season combinations were compared with Tukey's HSD test.

Differences between experimental groups were assessed using Student's *t* test or one-way analysis of variance (ANOVA) followed by Tukey's HSD post hoc test. Two-tailed one-sample *t* test was applied to compare the proportion of ET rate and WUE.

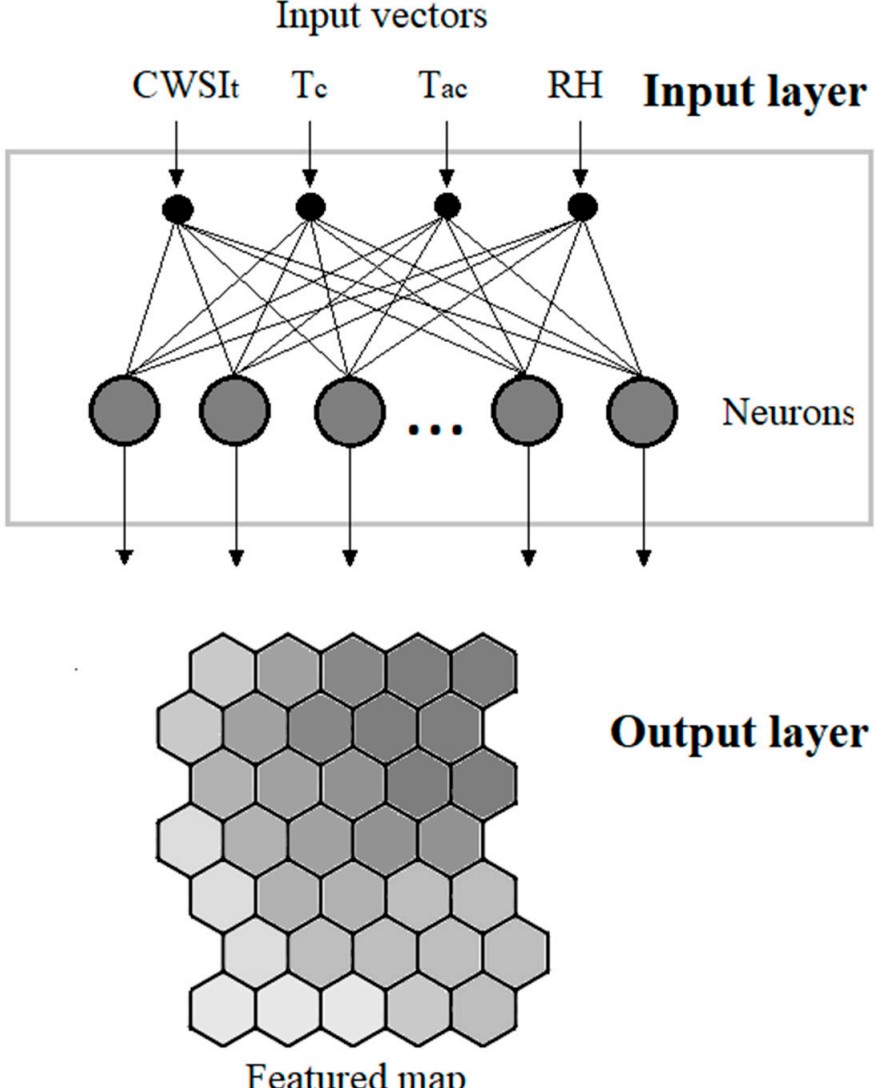

**Figure 1.** Schematic representation of the input and output vectors in the self-organization map K-SOM.

Regression analysis was applied to determine the relationship between ET, seed yield and $CWSI_t$. The goodness of curve fit was controlled by standard statistics as the $R^2$ and RMSE.

To facilitate the presentation of the results, 95% confidence interval was calculated.

Boxplots were applied to show differences between the daily ET rates, WUE and CWSI of soybean varieties, water treatments and seasons.

All statistical computations were implemented using SPSS software, version 24.0.

To investigate the performances for the model, different error measures including the root mean square error (*RMSE*), mean absolute error (*MAE*), coefficient of determination ($R^2$), scatter index (*SI*) and Nash-Sutcliffe efficiency (*NSE*) were used:

$$RMSE = \sqrt{\frac{\sum_{i=1}^{n}\left(CWSI_t - CWSI_p\right)^2}{n}} \tag{4}$$

$$MAE = \frac{\sum_{i=1}^{n}\left|CWSI_p - CWSI_t\right|}{n} \tag{5}$$

$$R^2 = \frac{\left(\sum_{i=1}^{n}\left(CWSI_t - \overline{CWSI_t}\right)\left(CWSI_p - \overline{CWSI_p}\right)\right)^2}{\sum_{i=1}^{n}\left(CWSI_t - \overline{CWSI_t}\right)^2 \sum_{i=1}^{n}\left(CWSI_p - \overline{CWSI_p}\right)^2} \qquad (6)$$

$$NSE = 1 - \frac{\sum_{i=1}^{n}\left(CWSI_t - CWSI_p\right)^2}{\sum_{i=1}^{n}\left(CWSI_t - CWSI_{pm}\right)^2} \qquad (7)$$

$$SI = \frac{RMSE}{CWSI_{pm}} \qquad (8)$$

where $CWSI_t$, $CWSI_p$ theoretical and predicted $CWSI$ values on the ith day, $CWSI_{pm}$ was the mean value of $CWSI_p$. The total number of testing patterns was denoted by $n$ and $i$ represent the number of particular instances of the testing pattern.

The K-SOMs were developed in MATLAB 2019b using the SOM toolbox.

## 3. Results

### 3.1. Weather Conditions between 2017–2020

Keszthely, located in the Carpathian Basin, is expected to have variable inter- and intra-annual weather conditions. The long-term seasonal total PR of 384.3 mm (1971–2000) can be typified by highly variable and irregular seasonal PR events ranging from 290.1 to 680.6 mm. Of the four growing seasons investigated, two wet (2018 and 2020) and two dry ones (2017 and 2019) could be distinguished (Table 1). 2017 and 2019 received 36.9 mm and 13.7 mm less PR, respectively, than the long-term average. Adding to the low seasonal PR in 2017, more than 40% of it fell in September, out of the soybean's growing period. The growing seasons of 2018 (+93.4 mm) and 2020 (+31.7 mm) had positive rainfall anomalies compared to climate norms. However, reduced, monthly PR totals during the 2017 growing season were evenly distributed throughout the soybean season. In wet 2018 and 2020, rainfall events were mostly concentrated between June and August, decreasing the monthly vapor pressure deficits (VPD). In the warm summer of 2017, the highest VPD values were observed between June and August. The seasonal mean Ta during the years 2017–2020 were 0.7–2.5 °C higher than the 30-yr average, except for April in 2017, and May in 2019 and 2020. Cooler springs 2017 and 2019 postponed the soybean seeding to the end of May. Annual mean Ta in the wet 2020 tended to be the lowest among the studied seasons.

**Table 1.** Meteorological variables, monthly mean temperatures, relative humidity, vapor pressure deficits and monthly precipitation sums measured at Agrometeorological Research Station of Keszthely in the growing seasons 2017–2020. The climate norm represented the period between 1971 and 2000.

| | Apr | May | June | July | August | September | |
|---|---|---|---|---|---|---|---|
| Precipitation sums [mm] | | | | | | | Total |
| Norm | 50.5 | 59.6 | 78.5 | 73.5 | 65.1 | 57.1 | 384.30 |
| 2017 | 20.9 | 38.8 | 61.1 | 53.8 | 32.7 | 140.1 | 347.40 |
| 2018 | 13.4 | 68.4 | 101.2 | 78.9 | 87.1 | 128.7 | 477.70 |
| 2019 | 28.7 | 125.0 | 50.4 | 92.1 | 25.9 | 48.5 | 370.60 |
| 2020 | 14 | 45.6 | 93.0 | 81.9 | 152.4 | 29.1 | 416.00 |
| Monthly mean air temperatures [°C] | | | | | | | Mean |
| Norm | 10.5 | 15.7 | 18.7 | 20.5 | 20.1 | 15.7 | 16.87 |
| 2017 | 10.8 | 16.6 | 21.2 | 22.3 | 22.8 | 15.1 | 18.14 |
| 2018 | 15.3 | 18.8 | 20.5 | 21.7 | 22.6 | 16.9 | 19.32 |
| 2019 | 11.9 | 13.0 | 22.8 | 22.8 | 22.6 | 17.1 | 18.39 |
| 2020 | 11.8 | 14.4 | 19.3 | 21.1 | 21.8 | 17.1 | 17.59 |
| Relative humidity [%] | | | | | | | Mean |
| 2017 | 67.4 | 71.0 | 70.1 | 68.3 | 66.8 | 81.9 | 70.9 |
| 2018 | 69.0 | 73.5 | 74.2 | 71.7 | 74.3 | 79.8 | 73.8 |
| 2019 | 70.1 | 80.2 | 74.1 | 69 | 76.9 | 77.8 | 74.7 |
| 2020 | 56.4 | 66.0 | 74.0 | 69.8 | 75.9 | 75.8 | 69.6 |

**Table 1.** *Cont.*

|  | Apr | May | June | July | August | September |  |
|---|---|---|---|---|---|---|---|
| Vapour pressure deficit [hPa] |  |  |  |  |  |  | Mean |
| Norm | 0.39 | 0.53 | 0.61 | 0.71 | 0.64 | 0.37 | 0.54 |
| 2017 | 0.43 | 0.57 | 0.76 | 0.87 | 0.94 | 0.33 | 0.65 |
| 2018 | 0.55 | 0.59 | 0.64 | 0.76 | 0.72 | 0.38 | 0.61 |
| 2019 | 0.44 | 0.31 | 0.73 | 0.87 | 0.64 | 0.43 | 0.57 |
| 2020 | 0.61 | 0.57 | 0.59 | 0.77 | 0.65 | 0.48 | 0.61 |

### 3.2. Soybean Development, ET and WUE

Irrespective of the two weeks earlier seeding in 2018, soybean development regarding the length of vegetation periods and phenological phases was similar across all treatments. Soybean vegetation cycles ranged from 114 to 122 days. In accordance with results of Montoya et al. [37], the water stress during flowering shortened the length of the vegetation cycle by 4–6 days. Soybean leaf senescence acceleration has been documented under drought conditions together with photosynthesis and assimilates reduction by Ergo et al. [38] in Manfredi, Argentina (31° N), during the growing seasons 2015–2016.

Daily mean ET rates for each treatment from emergence to maturity between 2017 and 2020 are presented in Figure 2. Four-season daily mean ET rates were $4.30 \pm 0.94$ mm day$^{-1}$ for Sin WW, $4.22 \pm 0.63$ mm day$^{-1}$ for Sig WW, $1.53 \pm 1.21$ mm day$^{-1}$ for Sin RO and $1.58 \pm 0.64$ mm day$^{-1}$ for Sig RO. Seasonal patterns of daily mean ET were similar to each treatment for the observed growing seasons, however the daily ET rate values varied among different treatments. Irrespective of treatment, low daily ET rates ranging from 0.1 to about 2 mm day$^{-1}$ occurred during the vegetative stages (V). Maximum daily ET rates between 4.1 (Sin RO in 2020) and 10.9 (Sig WW in 2017) mm during July (day of year, DOY: 190–200) coincided with the highest $T_a$ and crop flowering as reported by Anapalli et al. [8]. Daily mean ET rates gradually decreased in below 1.0 mm toward the end of the Augusts.

Daily mean ET rates were influenced by season, S ($p < 0.001$), water, W ($p < 0.001$) and one of their interactions S $\times$ W ($p < 0.001$), with the exceptions, non-significant variety, V ($p = 0.409$), interactions of V $\times$ S ($p = 0.41$), W $\times$ V ($p = 0.528$) and S $\times$ V $\times$ W ($p = 0.807$) effects on daily ET rates. The non-significant V $\times$ S interaction revealed that the response of S on daily mean ET rates did not vary between varieties.

Total ET varied widely with treatments, from 291.9 (Sig RO) to 694.9 mm (Sin WW) in 2017, from 315.7 (Sig RO) to 615.6 mm (Sin WW) in 2018, from 328.8 (Sig RO) to 646.1 mm (Sig WW) in 2019, and from 205.5 (Sin RO) to 419.7 mm (Sig WW) in 2020. Total ET$_0$ values for 2017, 2018, 2019 and 2020 growing seasons were estimated as 731.4 mm, 717.4 mm, 700.6 mm, and 510.9 mm, respectively.

In dry and wet growing seasons, the WUE varied from $0.70 \pm 0.02$ (Sig WW for 2017) to $1.0 \pm 0.04$ kg m$^{-3}$ (Sin RO for 2017) and from $0.84 \pm 0.03$ (Sig WW for 2020) to $0.93 \pm 0.07$ kg m$^{-3}$ (Sin RO for 2020), respectively, Figure 3. Figure 3 contains the dry and wet seasons pooled. The highest and most stable WUEs were obtained from RO, irrespective of weather conditions. The lowest WUE values were observed in WW during 2017. The size of CWSI and $T_c$ were inversely related to the amount of water used.

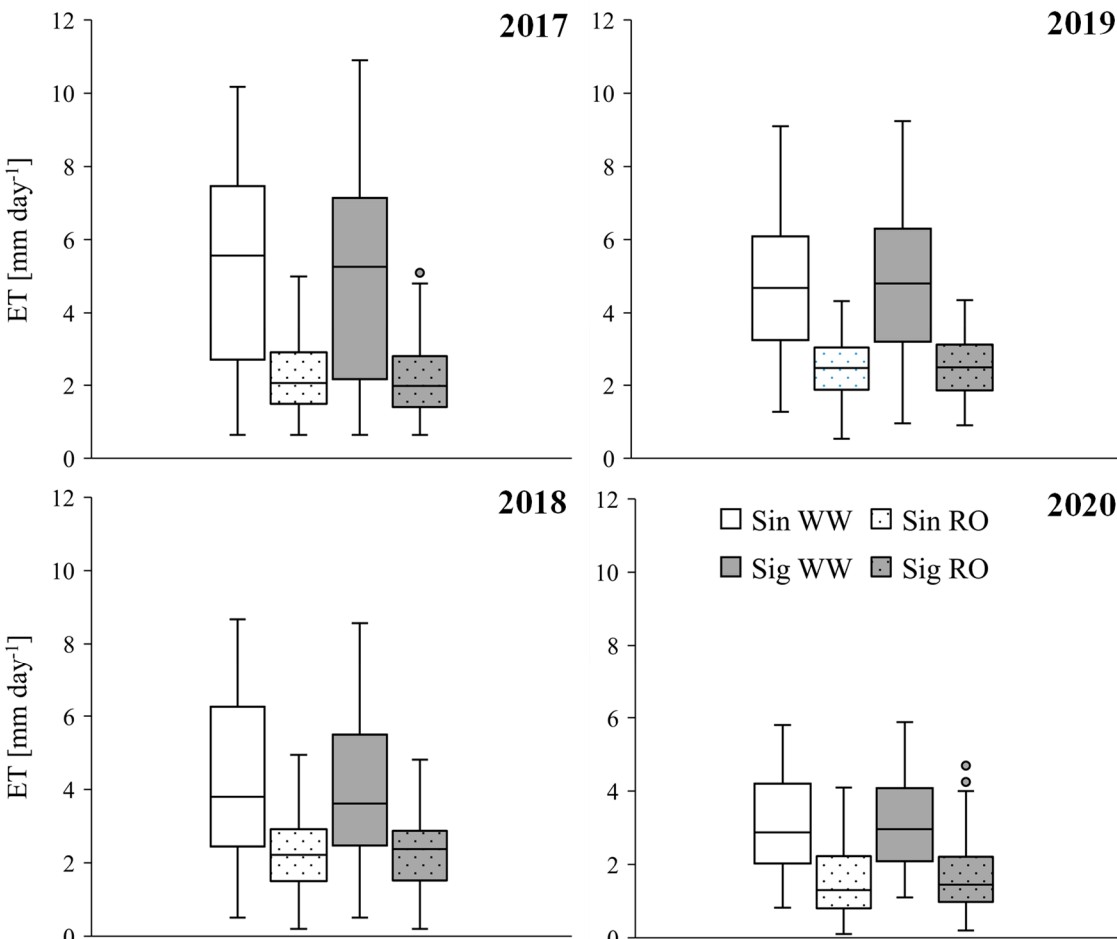

**Figure 2.** Daily mean evapotranspiration rates, ET [mm] under unlimited (WW) and water stress (RO) conditions in varieties Sinara (Sin) and Sigalia (Sig) during dry (upper row 2017 and 2019) and wet (lower row 2018 and 2020) growing seasons at Keszthely, Hungary.

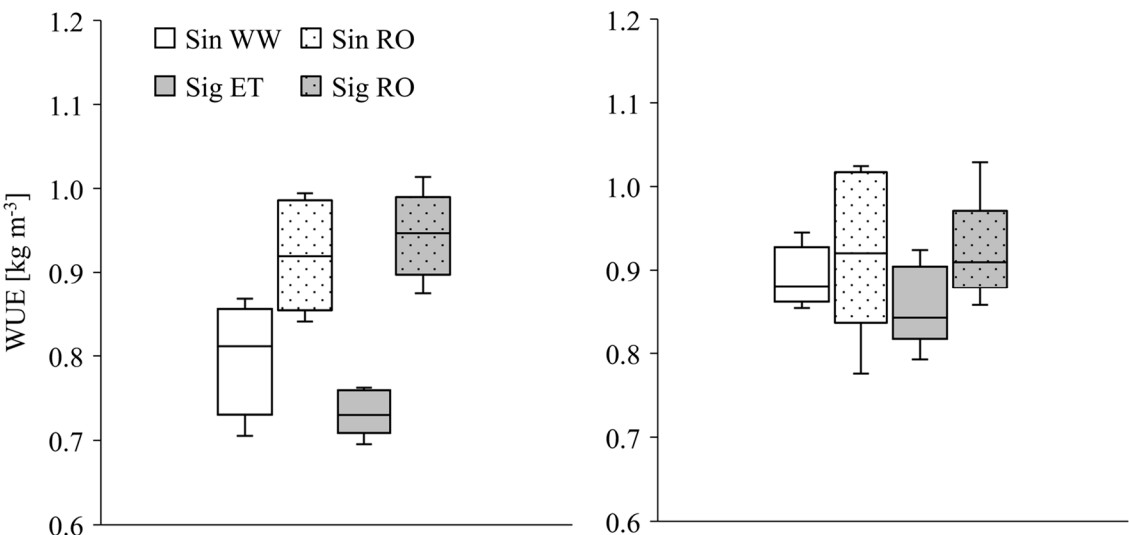

**Figure 3.** Water use efficiency (WUE) of soybean in unlimited (WW) and water stress (RO) conditions in pooled dry (2017 and 2019) and wet (2018 and 2020) growing seasons. The two soybean varieties used were Sinara, Sin and Sigalia, Sig.

Considering the ANOVA results, the WUE was affected by the W ($p < 0.001$) and one interaction of S $\times$ W ($p < 0.001$). In accordance with results in daily mean ET rates, the insignificance of variety, V ($p = 0.71$) and its interactions indicated that the effects of the S and W for WUE in the two studied varieties were similar.

*3.3. CWSI*

The CWSI values were calculated during the solar noon (12.00–13.30 LMT) consistently for all the treatments for three growing seasons (the fourth one was excluded from the analysis) when the canopy cover exceeded 70–80%, from the beginning of July to mid-August, to minimize the effect of soil temperature and change in canopy structure on the $T_c$ readings (Figure 4). Except for wet 2020, about every second day after canopy closure was appropriate for $T_c$ detection and computing CWSI during the three-season observation period (2017: 32 out of 66 days, 2018: 27 out of 55 days and 2019: 28 out of 57 days).

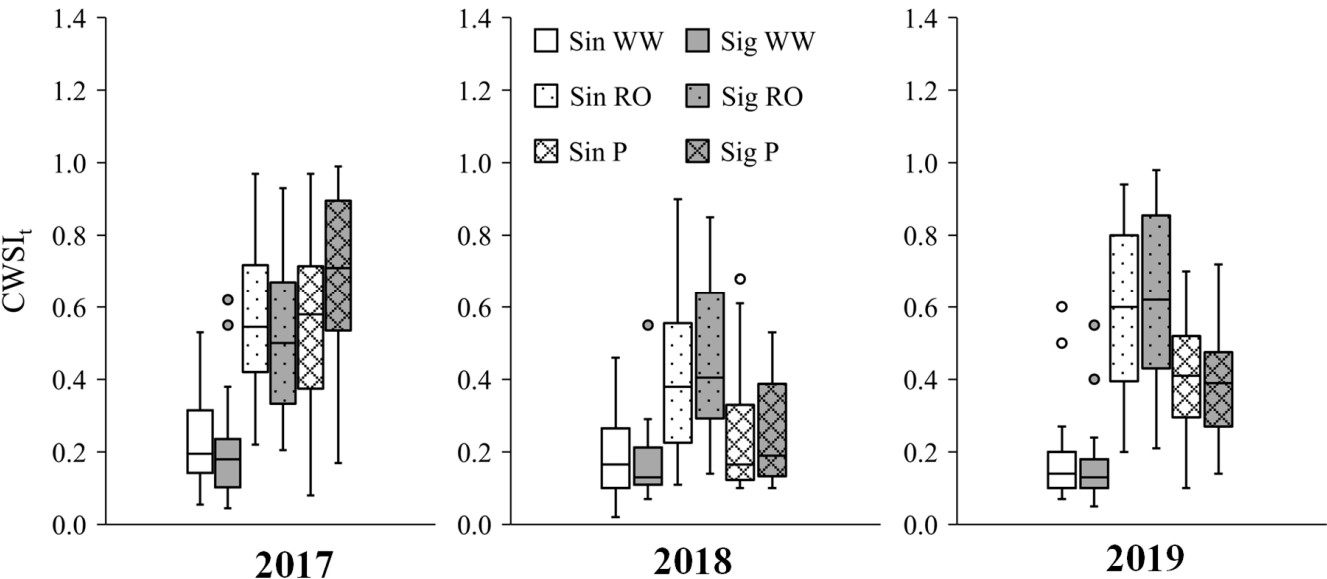

**Figure 4.** Seasonal mean of theoretical crop water stress index, CWSI$_t$ for soybean varieties Sinara (Sin) and Sigalia (Sig) during the seasons between 2017 and 2019. WW, RO and P denoted unlimited, water stress and rainfed, respectively.

Seasonal mean CWSIs of WW were close to each other ranging from $0.13 \pm 0.01$ (Sin in 2018) to $0.24 \pm 0.01$ (Sin in 2017) (Table 2). Water stress during flowering significantly increased the seasonal mean CWSI values from $0.36 \pm 0.20$ (Sin RO, 2018) to $0.64 \pm 0.02$ (Sig RO, 2019). CWSI values from $0.20 \pm 0.19$ to $0.59 \pm 0.11$ in Sin P (2018) and Sig P (2017), respectively, were ranging between well-watered and water-stressed soybean. Differences in CWSI values between variable water supplies were easily noticed. However, there was no difference in CWSI between the two varieties (their values were pooled in the analysis). High PR amount during wet 2018 might be high enough to allow almost optimum water supply, even in the rainfed. CWSI values decreased with decreasing $T_c$.

**Table 2.** Seasonal mean canopy temperature ($T_c$), air temperatures 1.5 m above the canopy ($T_{ac}$), canopy- air temperature difference ($T_c$-$T_{ac}$) and theoretical crop water stress index, $CWSI_t$ in the growing seasons of 2017–2019. Abbreviations were as follows: WW—unlimited water, RO—water stress conditions, P—rainfed, Sinara—Sin, Sigalia—Sig.

|  | $T_c$ | $T_a$ | ($T_c$-$T_{ac}$) | $CWSI_t$ |
|---|---|---|---|---|
| **2017** | | | | |
| Sin WW | 28.5 ± 2.27 | 28.5 | 0.0 | 0.24 ± 0.01 |
| Sig WW | 28.3 ± 2.21 | 29.2 | −0.9 | 0.20 ± 0.07 |
| Sin RO | 31.2 ± 3.14 | 29.0 | 2.2 | 0.58 ± 0.09 |
| Sig RO | 30.7 ± 2.29 | 29.5 | 1.2 | 0.52 ± 0.10 |
| Sin P | 31.1 ± 3.35 | 30.1 | 1.0 | 0.55 ± 0.19 |
| Sig P | 32.0 ± 3.20 | 30.2 | 1.8 | 0.59 ± 0.11 |
| **2018** | | | | |
| Sin WW | 27.7 ± 2.07 | 28.2 | −0.5 | 0.13 ± 0.01 |
| Sig WW | 28.3 ± 2.65 | 28.4 | −0.1 | 0.21 ± 0.01 |
| Sin RO | 29.4 ± 3.42 | 28.3 | 1.1 | 0.36 ± 0.20 |
| Sig RO | 29.5 ± 3.01 | 28.5 | 1.0 | 0.40 ± 0.20 |
| Sin P | 28.3 ± 2.66 | 28.9 | −0.6 | 0.20 ± 0.19 |
| Sig P | 28.3 ± 2.79 | 28.8 | −0.5 | 0.21 ± 0.14 |
| **2019** | | | | |
| Sin WW | 27.8 ± 2.10 | 28.0 | −0.2 | 0.17 ± 0.01 |
| Sig WW | 27.7 ± 2.08 | 28.2 | −0.5 | 0.16 ± 0.01 |
| Sin RO | 30.5 ± 2.71 | 28.4 | 2.1 | 0.61 ± 0.02 |
| Sig RO | 30.2 ± 2.88 | 28.7 | 1.5 | 0.64 ± 0.02 |
| Sin P | 29.1 ± 2.55 | 28.7 | 0.4 | 0.41 ± 0.16 |
| Sig P | 28.8 ± 2.50 | 28.3 | 0.5 | 0.39 ± 0.14 |

Three-season mean CWSI was influenced by season, S ($p < 0.001$), water, W ($p < 0.001$) and their interaction S × W ($p < 0.001$). The non-significant V ($p = 0.08$) interactions of V × W ($p = 0.39$), V × S ($p = 0.10$) and W × V × S ($p = 0.86$) on the CWSI revealed that the response of the seasons and water levels did not differ among varieties.

The application of K-SOM in CWSI computation was based on accessible meteorological variables that may contribute to a more accurate estimate of crop-water relationship and/or irrigation timing. As was expected, the ANOVA results clearly show the CWSI differences in the three water treatments, providing a wide range in K-SOM estimation. Low CWSI (CWSI < 0.15–0.20) was mainly related to WW, irrespective of season. High values of CWSI (~0.4–0.5) corresponded to water-stressed crops, which probability could be high in rainfed in Hungary. CWSI can be estimated by monitoring and using the easily observable meteorological elements ($T_{ac}$ and $RH_c$), in addition to $T_c$ sensing, to help professionals in agricultural practice. Thus, farmers with less meteorological information can take steps to address the negative impacts of drought in order to improve soybean yield and its stability.

The K-SOM, by using linear and non-linear data, was applied to evaluate whether the pattern recognition capacity could be improved through validation obtained from the data set. The quantization error (QE) is the mean to the distances between each input vector and its BMU. The topographic error (TE) shows the proportion of all data vectors, to which the first- and second-best matching units (BMUs) are not adjacent [39]. The QE and TE do not have a default value, but the smaller the QE and TE, the better the model performance is. In the best case the values tend to zero. In the study, the low values of QE (0.034) and TE (0.593) were obtained with an output layer of 10 × 6 neurons (Table 3).

**Table 3.** Characteristics of trained Kohonen-SOM model.

| Characteristics | Values |
| --- | --- |
| Normalization method | variance : $x' = (x - \bar{x})/\sigma_x$ |
| Codebook | $160 \times 4$ |
| Map Size | $10 \times 6$ |
| Neighbourhood function | Gaussian |
| Shape | Sheet |
| Lattice | Hexagonal |
| Final Topographic error (TE) | 0.593 |
| Final Quantization error (QE) | 0.034 |

The component planes of the database show the correlation patterns between meteorological variables and CWSI. In Figure 5, each variable was presented by a grid of 60 neurons (maps) for different water treatments, with 6 vertical and 10 horizontal columns. The similar color patterns in the maps demonstrated positive correlation between the studied variables and CWSI. On the maps, the warm and cold colors indicate high and low values, respectively. The identical color patterns on the maps demonstrated a positive correlation between the study parameters.

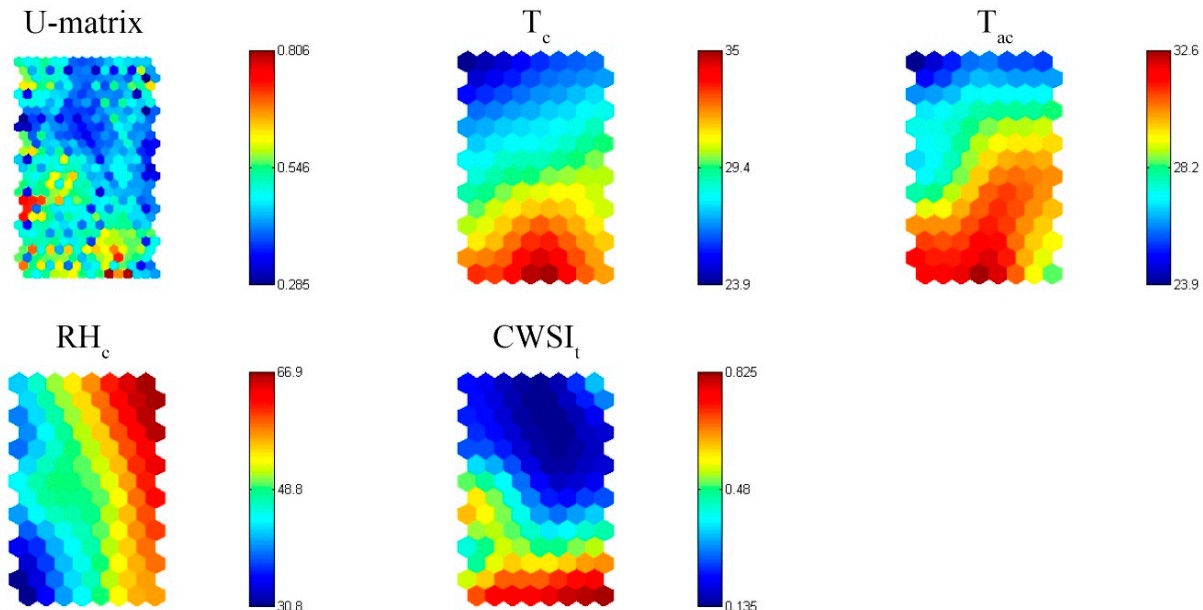

**Figure 5.** Component planes of the variables ($T_c$—canopy temperature, $T_{ac}$—air temperature above the canopy, $RH_c$—relative humidity above the canopy) for crop water stress index, $CWSI_t$.

The u-matrix (Figure 5) was a map, generated by the program, to show the relations between the neighboring neurons. A light color between the neurons signifies that the codebook vectors were close to each other in the input space, while dark colors meant large distances [40]. The color gradient of the $T_{ac}$ component plane is significantly correlated with the $T_c$ component plane, some deviation is found only in the plane on the right (Figure 5). The $RH_c$ component plane presents a negative correlation with the $T_{ac}$ and $T_c$ component planes, although it shows greater overlap with $T_{ac}$ (right plane) compared to $T_c$. CWSI correlates with all three meteorological variables; high $T_{ac}$ and $T_c$ and low $RH_c$, as well as medium $RH_c$ and high $T_{ac}$ values increase the CWSI value. Low $RH_c$ and $T_{ac}$ values can be correlated with low CWSI values.

## 4. Discussion

On several occasions in this study, on 50 days out of 62 (80.6%) throughout the sampling period 2020, there were inappropriate weather conditions in the $T_c$ data collection ($T_{max} \leq 25.0$ °C; global radiation $\leq 20$ MJ day$^{-1}$, daily mean windspeed $\geq 2.5$ m s$^{-1}$, and cloudiness about solar noon). Due to uncertain $T_c$ results in treatment comparisons even on the remaining sample days (19.4%), the $CWSI_t$ was disregarded in the analysis during wet 2020. Most of the days with inaccurate $T_c$ samplings coincided with decreases in global radiation and were associated with cloudy and windy weather conditions around solar noon. In a temperate humid climate, Jensen et al. [41] observed extreme fluctuations in $T_c$ attributed to temporal variability in global radiation, increased wind speed and to the narrow range of prevailing values of VPD. The authors mentioned that under conditions of low evaporative demand, $T_c$ differences between water stressed and fully irrigated crops approached zero even at severe crop water stress. The CWSI showed large fluctuations under low VPD or with significant variation in weather conditions [42], although Jackson et al. [11] found the stability of theoretical $CWSI_t$ advantageous in variable climate.

Under the constraint of declining PR due to the impacts of global warming and sustainable agriculture, irrigation planning must be developed in the direction of water saving and precision. Traditional irrigation planning methods are based on soil moisture data, meteorological variables, and crop indicators. ET is a multivariate process influenced by meteorological and biological variables. $T_c$, a biological indicator, was sensed remotely, which is a non-destructive, simple, and fast way to provide a rapid estimate of crop water supply level. Among meteorological variables involved in ET regulation, PR, solar radiation ($T_a$), humidity (VPD, RH) and windspeed (u) were found mostly intercorrelated [43]. The VPD, a critical weather variable, increases with increased $T_a$, and controls soybean ET via stomatal regulation [9]. Lower $T_a$ in April and May slightly increased initial ET rates. Due to higher $T_a$ and LAI values, the daily ET rates were characterized by rapid increases from June to about mid-August. Because of leaf senescence from the middle of August, mid-summer ET rates were followed by a gradual decrease towards September.

The lowest daily mean ET rates of 1.5 (Sin) and 1.6 (Sig) mm in RO, and 3.1 (Sin; Sig) mm in WW were measured during the wet 2020. The highest daily mean ET rate increments during the dry 2017 were 64.1% in Sin WW and 46.9% in Sig WW, compared to those from the wet 2020. During the dry 2017, the daily mean ET rates for RO were 42.1 and 31.6% higher in Sin and Sig, respectively, relative to daily mean ET rates during 2020.

During the dry growing seasons, the ET total of Sin WW (mean, 664.8 ± 2.26 mm) was similar to that of the ET total of Sig WW (mean, 651.8 ± 2.39 mm), hence there was no detectable variety effect on soybean ET sum with unlimited watering. Similarly, during those same arid growing seasons, the ET total of Sin RO (mean, 319.8 ± 1.72 mm) was not statistically distinguishable from that of Sig RO (mean, 310.3 ± 1.69 mm). During the wet seasons (2018 and 2020), we observed mean ET total decreases of 25.8% ($p < 0.001$) in Sin WW and 26.8% ($p < 0.001$) in Sig WW, as compared to the ET sums in arid weather. At the same time, 20.1% ($p < 0.001$; Sin RO) and 16.4% ($p < 0.001$; Sig RO) declines in four-season mean ET sums were obtained from water stressed crops. Cumulative ET changed seasonally and between water treatments within the same growing season. Seasonal variations in cumulative ET were mainly the result of changed weather conditions, water supply levels and biological factors. Among biological factors, the size of leaf-area plays an important role. In the previous results we concluded that 50% water withdrawal at the time of flowering significantly reduced the seasonal mean LAI of both varieties in comparison to WW [23,44]. Irrespective of variety and season, declines in seasonal mean LAI of RO were two-fold higher than that of the average LAI in rainfed [29]. More detailed LAI results including temporal variation and leaf-area modelling were excluded from the study; they were presented in the previous publications (see above). For all seasons, unlimited watering during flowering always produced 67.8 (Sin) and 66.9% (Sig) higher ET total compared to their water-stressed counterparts. Similar results have also been obtained by Payero et al. [45] in West-Nebraska (41.4° N) for soybean ET at not limited soil

water (630–641 mm) and water constraint restricted to reproductive development (310 mm). Three-season mean ET totals of 640 and 389 mm for full and limitedly irrigated soybean in central Nebraska reported by Schneekloth et al. [46] are also comparable to the results in the study. Seasonal totals also showed that wet seasons had narrower range, while dry seasons had wider range in cumulative ET, irrespective of variety. In humid seasons, the difference in ET between the fully irrigated and deficit irrigated soybean was also the smallest in Vojvodina, Serbia (44° N), in a three-season study [47]. The authors concluded that in wet seasons, no irrigation of soybean was required in the temperate environment of Serbia.

For the whole investigation period, ET totals of WW were 78.5 (humid seasons mean) and 91.2% (dry seasons mean) of $ET_0$. The ET sums of water stressed crops were 55.9 and 57.3% of $ET_0$ for 2018 and 2020, and 2017 and 2019, respectively.

Regression analysis between the soybean yield, y [kg m$^{-2}$] and ET total [m] showed a quadratic relationship (y = $-0.0042 \times 2 + 4.573 - 763.43$, $R^2$ =0.9398, $p < 0.001$, RMSE = 0.066 kg m$^{-3}$) for combined data from 2017, 2018, 2019 and 2020. Therefore, optimum water uses of 544.4 mm existed at the study site to maximize the yield. A somewhat lower ET total of 380 mm was observed by Gajic et al. [47] in Serbia, to guarantee high yield in soybean. Candogan et al. [48] in Turkey (40° N) reported the same quadratic relationship between ET and soybean yield, in a semi-humid environment. In contrast, many studies demonstrated a linear relationship between ET and soybean yield, most of them under arid or semi-arid weather conditions (Payero et al. [45] and Kirnak et al. [49] both in Nebraska (41° N).

Considering previous observations of Montoya et al. [37], the ET sums and WUE were strongly correlated. A previous study of Henry et al. [50] in Stuttgart, Germany, reported that as water becomes limited, WUE improves. When varieties were pooled, during the dry and wet growing seasons, 15.7 ($p = 0.016$) and 7.0% ($p = 0.017$) increments in WUE of RO, respectively, were determined relative to WW. WUE values reported in the study were comparable to those determined by Hussain et al. [51] for rainfed (0.7 kg ha$^{-1}$ mm$^{-1}$) and water stressed soybean (0.46 kg ha$^{-1}$ mm$^{-1}$) in southwest Michigan, USA (42° N), between 2009 and 2012. Higher WUE in the water-stressed soybean were likely associated with a decline in photosynthesis due to water withdraw, evident from the increased $T_c$ and higher CWSI observed in the RO treatments. Photosynthesis, a process that contributes to crop growth, development, as well as soybean yield is highly sensitive to water stress [52]. Wijewardana et al. [53] concluded that regardless of soybean cultivar, the decrease in photosynthesis is mainly due to stomatal closure. No significant differences in WUE of RO between varieties were found over the four-season study. During both dry and wet seasons, the WUEs in RO were similar to those in rainfed ones (wet: $p = 0.845$; dry: $p = 0.114$). An 18.1% increase ($p < 0.001$) in the WUE of P was only observed in dry summer, relative to values in WW. We expected that the WUE of WW during arid seasons would be greater than those in wet seasons. Contrary to our expectations, the WUE values decreased by 7.2% ($p = 0.011$) during arid seasons compared to those in the wet ones. As with WUE-seed yield, variation in ET rather than yield explained most of the variation in WUE. The SDs of WUE were always much lower in WW than in RO and P, which means that the additional water balanced the WUE in case of unlimited watering, irrespective of the weather conditions.

$CWSI_t$ values were strongly influenced by different water supplies. In dry growing seasons, mean $CWSI_t$ of RO and P reflected the increased water stress compared to stress values of WW. It can be concluded that there was a negative correlation between the total ET and seasonal mean ($T_c$-$T_{ac}$), that is, the ET decreased with the increase in ($T_c$-$T_{ac}$) with different watering levels (see also Table 2). With unlimited water supply and in P during the wet 2018, the seasonal mean ($T_c$-$T_{ac}$) values were negative, and the variations in both $CWSI_t$ and $T_c$ were lower while ET was higher. Han et al. [42] commented that ET reduces $T_c$ and ($T_c$-$T_{ac}$); and lower $T_c$ maintains a green character of soybean and improves chlorophyll retention, a mechanism to avoid crop drying, which contributes to higher photosynthetic rate and more dry matter accumulation, producing higher yield. In addition to absolute $T_c$

readings, the SD of $T_c$, as a derived index, could also be applied as a canopy indicator in selecting better varieties to water stress tolerance.

This study attempted to identify the impact of $CWSI_t$ on soybean yield under well-watered, rainfed and water stress conditions. As the $CWSI_t$ at solar noon during the flowering and podding stages were stable and most relevant to soybean grain yield [42], the calculated mean CWSIs from flowering were fitted with corresponding yields over the seasons 2017–2019 (Figure 6). The three-season average $CWSI_t$ of 0.19 for WW produced the highest seed yield. Similar CWSI values of 0.17 and 0.22 (2-year-study) have also been reported for fully irrigated soybean by Candogan et al. [48] in Turkey (40° N). CWSI concurred with a value of 0.2 for fully irrigated soybean, published by Nielsen [54] in the Central Great Plains Research Station, USA (41° N). Over the three-season study, water stress under flowering and rainfed conditions increased the $CWSI_t$ (RO = 0.52 ± 0.11; $p$ = 0.39 ± 0.16), reducing yield by about half. However, there was hardly enough variation in $CWSI_t$ between rainfed and WW during the wet 2018 (see also Figure 4).

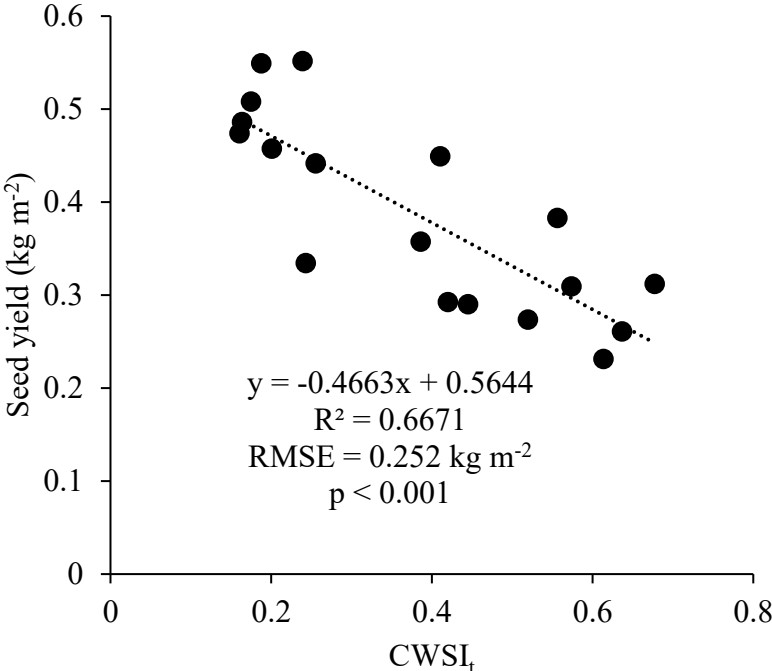

**Figure 6.** Relationship between theoretical crop water stress index, CWSIt and soybean seed yield of two varieties, Sinara (Sin) and Sigalia (Sig) grown under three watering levels (unlimited, water withdrawn and rainfed) during the seasons between 2017–2019. A single linear function was used to describe the relationship.

The larger the amount of available water to answer crop water demand, the higher the expectable soybean grain yield. In accordance with the results of Singh et al. [55] in Nebraska (41° N), for the period of 2018 and 2019, a linear equation described the relationship between seasonal mean CWSI and average yield [kg m$^{-2}$] in soybean (y = $-0.4663$x + 0.5644; $R^2$ = 0.6671; RMSE = 0.252; $p$ < 0.001). In this study, each 0.1 increase in CWSI above 0.19 at about solar noon would on average reduce the soybean grain yield by about 0.50 kg m$^{-2}$ in 2017–2019, which is 9.9, 18.0 and 13.1% of the yield for WW, RO, and P, respectively, when combining data for all seasons. The $CWSI_t$ could explain 74.4% of the variation in yield over the study period.

Therefore, this study concluded that reduced water received during soybean flowering before yield components were established resulted in lower soybean yield. Similarly, Wei et al. [56] discovered by pot experiment that drought at the flowering stage resulted in a larger yield loss in soybean. Furthermore, Pajero and Irmak [30] concluded that more water

provided early enough during the soybean vegetative period, well before yield components were completed, also could produce higher grain yield.

The performance of the K-SOM model is computed through error statistics and X–Y scatter plots (Figure 7). The error statistics of K-SOM model during testing (2019) and training (2017 and 2018) periods are presented in Table 4. Error statistics indicated accurate performance of the K-SOM model in predicting the CWSI for soybean.

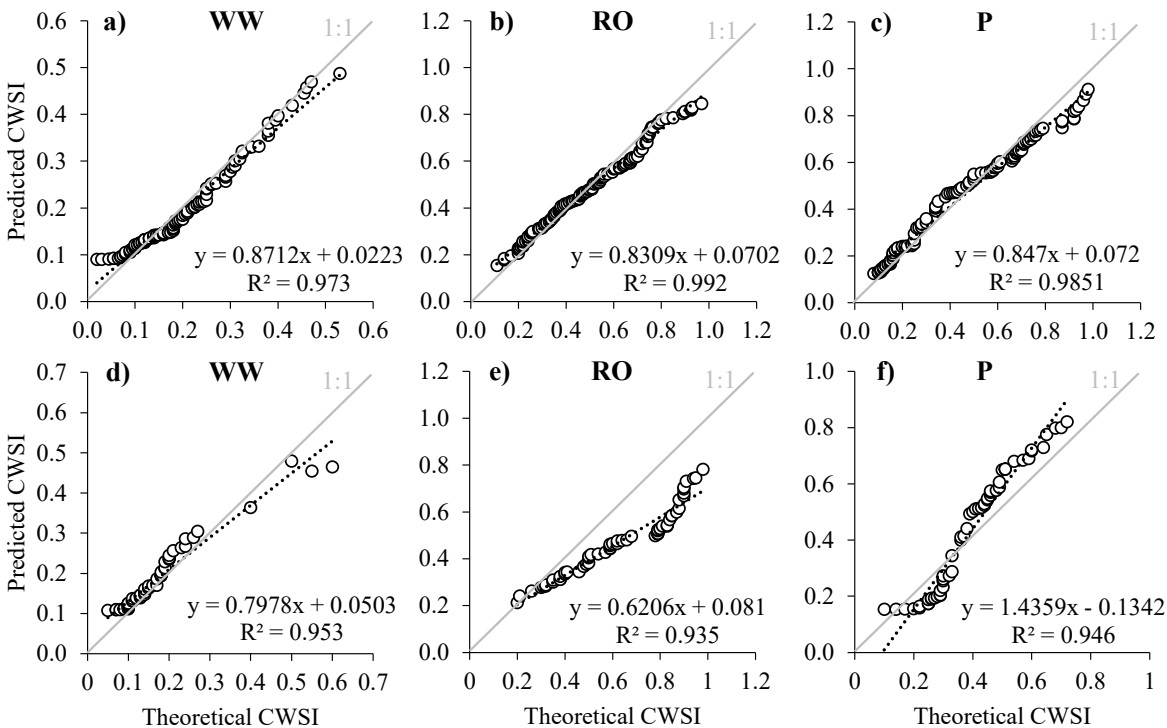

**Figure 7.** Comparison of theoretical and predicted CWSI values for the KSOM in the training and testing periods from (**a**–**c**) and from (**d**–**f**), respectively.

**Table 4.** Testing statistics of KSOM model. The coefficient of determination ($R^2$), root mean square error (RMSE), scatter index (SI), Nash–Sutcliffe (NSE) of CWSI predictions.

| | $R^2$ | RMSE [mm] | MAE | NSE | SI |
|---|---|---|---|---|---|
| Training period (2017–2018) | | | | | |
| WW | 0.973 | 0.019 | 0.017 | 0.961 | 0.101 |
| RO | 0.992 | 0.032 | 0.033 | 0.961 | 0.067 |
| P | 0.985 | 0.047 | 0.043 | 0.965 | 0.105 |
| Testing period (2019) | | | | | |
| WW | 0.953 | 0.033 | 0.026 | 0.901 | 0.182 |
| RO | 0.935 | 0.046 | 0.158 | 0.336 | 0.095 |
| P | 0.946 | 0.068 | 0.083 | 0.617 | 0.152 |

The correlation between theoretical $CWSI_t$ and K-SOM prediction was high, ranging from 0.935 to 0.992 during the model training and testing periods. Figure 7 showed the X–Y scatter plots of the linear regression between theoretical and predicted $CWSI_p$ which demonstrated a uniform scatter around the 1:1 line. The prediction accuracy by data-driven K-SOM depends on its "learning" efficiency, the amount and range of used $CWSI_t$ during training period. Lack of available $CWSI_t$ in some weather cases during training period may produce "gaps" in the testing period (see Figure 7). Thus, performance of theoretical and predicted CWSI values could be improved with an increased amount of input data presenting wide range of $CWSI_t$ (different weather conditions). If missing $CWSI_t$ data in the "gaps" were available, probably the K-SOM could learn more from them, improving the

model projection accuracy. Table 4 also demonstrated the test statistics of each K-SOM in terms of $R^2$, RMSE, SI, and NS for all water treatments, in training (2017–2018) and testing periods (2019). High $R^2$ and low RMSE values indicated a close relationship between data sets. Low values were observed for MAE, except for RO treatment in the testing period. NSE values close to 1 were observed during the training period, regardless of water supply. During the testing period, only WW treatment had a high NSE value, and RO and P treatment showed a significant decrease. SI recorded the lowest values for RO treatment in both the testing and training periods.

K-SOM method carried the advantage of studying the $CWSI_p$ identification, through plane-plots and generated maps, in addition to interpreting non-linear relationships. The aim of this study was the investigation of soybean-water relationship based on $CWSI_p$ determination. Further analysis of K-SOM-based $CWSI_p$ including yield parameters, soil moisture depletion and WUE can provide accurate information for effective irrigation scheduling.

## 5. Conclusions

To prepare a comprehensive irrigation decision, three different watering levels were designed (rainfed, unlimited, and water stress in flowering) focusing on the impact of water withdrawal on ET and $CWSI_t$ during soybean flowering. The meteorological elements completed with $T_c$ data provided a basis in CWSI estimation. As the applicability of $T_c$-based $CWSI_t$ is limited to certain "ideal" weather conditions such as calm and clear-sky solar radiation [56], the humid season of 2020 was excluded from the analysis. The $CWSI_t$ can be a useful tool for managing irrigation scheduling and water stress in the field, although in various weather conditions, mainly during humid seasons, must be handled with care when attempting to detect $T_c$.

This four-season investigation indicated that the two studied soybean varieties with varied water demands hardly changed in their response to water stress during flowering, as no modification connected to water requirement occurred.

If the provided water amount rose over 450 mm during the growing season, only the ET increased but not the soybean yield. Soybean seed yield and PR results showed that seasonal PR sums (2017: 347.4 mm and 2019: 370.6 mm) during the dry growing seasons might be insufficient to achieve high seed yield. The long-term PR total (384 mm) is also lower than that of the estimated soybean water demand based on $CWSI_t$ vs. ET relationship, at Keszthely. During dry seasons the seed yield would be increased when up to 100–160 mm of irrigation water is added. The results in this study in the temperate climate of Hungary suggest that the use of $CWSI_t$ to schedule irrigation may only be recommended during the arid growing seasons. In humid ones several limitations of $T_c$ sensing would prevent accurate $CWSI_t$ calculation and proper irrigation timing.

A unique neural network methodology, the Kohonen self-organizing map was applied to visualize the $CWSI_p$ and its components. The training epochs and $6 \times 10$ topology proved adequate for the separation of easily accessible meteorological elements impacting the CWSI. The theoretical $CWSI_t$ and the K-SOM predicted one showed strong agreement in the range 0.19–0.64, having $R^2$ of at least 0. 934, irrespective of variety. This result is promising, as the local rainfed values are inside the 0.2–0.6 index-range. The varieties were considered stable in every watering level, without performance variation in the growing seasons and watering levels.

**Author Contributions:** Conceptualization, A.A. and G.S.; methodology, G.S.; software, B.S.-G..; validation, A.A., B.S.-G. and G.S.; formal analysis, B.S.-G.; investigation, G.S.; data curation, B.S.-G.; writing—original draft preparation, A.A.; writing—review and editing, A.A., B.S.-G. and G.S.; visualization, B.S.-G.; supervision, A.A., B.S.-G. and G.S. All authors have read and agreed to the published version of the manuscript.

**Funding:** This research received no external funding.

**Institutional Review Board Statement:** Not applicable.

**Informed Consent Statement:** Not applicable.

**Data Availability Statement:** The data presented in this study are available on request from the corresponding author.

**Acknowledgments:** The publication is supported by the EFOP-3.6.3-VEKOP-16-2017-00008 project. The project is co-financed by the European Union and the European Social Fund. A special thanks to Karintia Ltd. for their kindness by supporting us by donating good-quality, disease-free soybean seed.

**Conflicts of Interest:** The authors declare no conflict of interest.

## Abbreviations

The following abbreviations are used in the manuscript:

| | |
|---|---|
| ANOVA | one-way analysis of variance |
| ARS | Agrometeorological Research Station |
| BMU | Best Matching Unit |
| Cfb | Köppen climate classification: oceanic climate with war summers |
| $c_p$ | heat capacity of air in constant pressure, $J\,kg^{-1}\,K^{-1}$ |
| CWSI | Crop Water Stress Index |
| $CWSI_p$ | predicted CWSI |
| $CWSI_{pm}$ | the mean value of $CWSI_p$ |
| $CWSI_t$ | theoretical CWSI of Jackson et al. |
| DOY | Day-Of-Year |
| $e_s(T_c)$–e | the difference in saturation and actual vapor concentrations of air, hPa |
| ET | evapotranspiration, mm day$^{-1}$ |
| $ET_0$ | FAO-56 Penman-Monteith reference evapotranspiration, mm day$^{-1}$ |
| ha | acre, 10,000 m$^2$ |
| HSD | Honestly Significant Difference |
| K-SOM | Kohonen Self-Organizing Maps |
| LAI | Leaf Area Index, $m^2\,m^{-2}$ |
| MAE | Mean Absolute Error |
| NSE | Nash-Sutcliffe Efficiency |
| $p$ | air density, kg m$^{-3}$ |
| p | probability |
| P | rainfed field treatment, mm day$^{-1}$ |
| PR | precipitation, mm day$^{-1}$ |
| QE | Quantization Error |
| R1 | beginning bloom stage |
| $R^2$ | coefficient of determination |
| R4-R5 | grain-filling stage |
| $r_a$ | aerodynamic resistance, s m$^{-1}$ |
| $RH_c$ | relative humidity above canopy, % |
| RMSE | Root Mean Square Error |
| $R_n$ | net radiation, W m$^{-2}$ |
| RO | water stressed treatment, mm day$^{-1}$ |
| S | season |
| SD | Standard Deviation |
| SI | Scatter Index |
| Sig | Sigalia (soybean var.) |
| Sin | Sinara (soybean var.) |
| $T_a$ | seasonal mean temperature, °C |
| $T_{ac}$ | temperature above canopy, °C |
| $T_{ac}$ | the ambient air temperature, °C |
| $T_c$ | canopy temperature, °C |
| $T_c$-$T_a$ | canopy- air temperature difference, °C |
| TE | Topographic Error |
| UV | ultraviolet radiation, 10 nm–400 nm |

| V | variety |
| VPD | Vapor Pressure Deficit, hPa |
| W | water |
| WUE | Water-Use Efficiency, kg m$^{-3}$ |
| WW | unlimited water supply treatment, mm day$^{-1}$ |
| y | yield, kg m$^{-2}$ |
| $\gamma$ | psychrometric constant, hPa K$^{-1}$ |
| $\Delta$ | the slope of saturated vapor pressure-temperature relation, hPa K$^{-1}$. |

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
