# Peer review of "The Application of a Self-Organizing Model for the Estimation of Crop Water Stress Index (CWSI) in Soybean with Different Watering Levels"

_water, doi:10.3390/w13223306_

Round 1
Reviewer 1 Report
Dear Authors
Using K-SOM (Kohonen self-organizing maps) algorithm to predict CWSI is very interesting article because water is inevitable component in soybean from the protection drought.
However, reviewer has some major and minor comments, please check it below:
1. Please check the abstract according to journal guideline.
2. Numerous models are revised and algorithms are generated from regression fitting to data by using different equations; but, not model improvement because parameter values in any equations are context dependent. In this connection, what was the actual improvement of K-SOM model for CWSI in this manuscript; instead, an ensemble of previous K-SOM model has been proposed by other authors? Please clarify it on the manuscript. Otherwise, in future, crop water stress index (CWSI) has just the K-SOM model to predict, but K-SOM model understand the less certain in the biological meaning of crop and other conditions.
3. In figure 7 (d, e, f), there are some gap between predicted and theoretical CWSI value in 1:1 Lin during the testing period. Authors should clarify and explain it.
Thank you!
Author Response
Using K-SOM (Kohonen self-organizing maps) algorithm to predict CWSI is very interesting article because water is inevitable component in soybean from the protection drought.
However, reviewer has some major and minor comments, please check it below:
- Please check the abstract according to journal guideline.
The control of the Abstract was carried out in accordance with the Journal’s guidelines. Abbreviations of different CWSI forms were corrected and the citation was omitted from the text.
- Numerous models are revised and algorithms are generated from regression fitting to data by using different equations; but, not model improvement because parameter values in any equations are context dependent. In this connection, what was the actual improvement of K-SOM model for CWSI in this manuscript; instead, an ensemble of previous K-SOM model has been proposed by other authors? Please clarify it on the manuscript. Otherwise, in future, crop water stress index (CWSI) has just the K-SOM model to predict, but K-SOM model understand the less certain in the biological meaning of crop and other conditions.
The aims of this manuscript were clearly identified as follows: 1) the extension of the previous findings for CWSI under highly variable weather conditions of Hungary, 2) applying easily accessible meteorological and crop variables to get CWSI 3) analysing ET and CWSI of two soybean varieties differing in their water demands. The novelty of the paper was also mentioned below the aims; the used meteorological and plant variables are easily accessible to future users at any meteorological station.”
We added to the Introduction accepting the reviewer’s notice: “Although the biological meaning of crops may be certain using K-SOM projected CWSIp only. K-SOM projected CWSIp may provide an alternative to other CWSI estimations requiring large amount of input data and computation.”
- In figure 7 (d, e, f), there are some gap between predicted and theoretical CWSI value in 1:1 Lin during the testing period. Authors should clarify and explain it.
The prediction accuracy by data-driven K-SOM depends on its “learning” efficiency, the amount and range of used CWSIt during training period. Lack of available CWSIt in some weather cases during training period may produce “gaps” in the testing period (see Fig. 7). Thus, performance of theoretical and predicted CWSI values could be improved with increased amount of input data presenting wide range of CWSIt (different weather conditions). If missing CWSIt data in the “gaps” were available, probably the K-SOM could learn more from them, improving the model projection accuracy.
Thank you!
Reviewer 2 Report
The paper was revised according to the journal rules. The topic treated deserves to be considered, especially in this age. The manuscript is well written with a good scientific point of view.
Only few revisions are required and they are reported below:
- a nomenclature section for all parameters and variables used in the paper should be' added. Add also proper SI unit of measure
- Check that all details for the istruments used are properly added to the manuscript
- In table 1 it should be better to show standard deviation values
- Add the proper reference for table 1
Author Response
- a nomenclature section for all parameters and variables used in the paper should be' added. Add also proper SI unit of measure
Abbreviations were added to manuscript
- Check that all details for the istruments used are properly added to the manuscript
Done
- In table 1 it should be better to show standard deviation values
No in case of weather variables. The climate norm is the basis that helps to characterise the weather conditions of a given month/season as it is in the Table.
- Add the proper reference for table 1
These data are our own measurements (under the provision of the Hung. Met. Service). Reference is not needed.
Thank you!
Reviewer 3 Report
It would be good to include some information about the importance of soybean as a food in vegetarianism in introduction.
Author Response
It would be good to include some information about the importance of soybean as a food in vegetarianism in introduction.
We added to introduction this: „…an important meat and dairy substitute for vegetarians”
Thank you!